# Multivariate Curve Resolution Alternating Least Squares Analysis of In Vivo Skin Raman Spectra

**DOI:** 10.3390/s22249588

**Published:** 2022-12-07

**Authors:** Irina Matveeva, Ivan Bratchenko, Yulia Khristoforova, Lyudmila Bratchenko, Alexander Moryatov, Sergey Kozlov, Oleg Kaganov, Valery Zakharov

**Affiliations:** 1Department of Laser and Biotechnical Systems, Samara University, Samara 443086, Russia; 2Department of Oncology, Samara State Medical University, Samara 443099, Russia; 3Department of Visual Localization Tumors, Samara Regional Clinical Oncology Dispensary, Samara 443031, Russia

**Keywords:** alternating least squares, benign neoplasm, malignant neoplasm, multivariate curve resolution, Raman probe, Raman spectroscopy

## Abstract

In recent years, Raman spectroscopy has been used to study biological tissues. However, the analysis of experimental Raman spectra is still challenging, since the Raman spectra of most biological tissue components overlap significantly and it is difficult to separate individual components. New methods of analysis are needed that would allow for the decomposition of Raman spectra into components and the evaluation of their contribution. The aim of our work is to study the possibilities of the multivariate curve resolution alternating least squares (MCR-ALS) method for the analysis of skin tissues in vivo. We investigated the Raman spectra of human skin recorded using a portable conventional Raman spectroscopy setup. The MCR-ALS analysis was performed for the Raman spectra of normal skin, keratosis, basal cell carcinoma, malignant melanoma, and pigmented nevus. We obtained spectral profiles corresponding to the contribution of the optical system and skin components: melanin, proteins, lipids, water, etc. The obtained results show that the multivariate curve resolution alternating least squares analysis can provide new information on the biochemical profiles of skin tissues. Such information may be used in medical diagnostics to analyze Raman spectra with a low signal-to-noise ratio, as well as in various fields of science and industry for preprocessing Raman spectra to remove parasitic components.

## 1. Introduction

In recent years, methods such as Raman spectroscopy have been increasingly used to investigate biological tissues and fluids [1,2,3]. The Raman spectra of various biological tissues are specific: during the development of a neoplasm, some biochemical changes occur, which leads to changes in the measured Raman spectra. Previously we showed that Raman spectroscopy can be used in the diagnosis of skin cancer [4], kidney-failure detection [5], and for blood analysis [6]. These studies demonstrate that Raman spectroscopy can be used to assume diagnostic conclusions, i.e., to determine a disease or tumor type.

The use of Raman spectroscopy in medical diagnostics is facilitated by the development of technical means for recording Raman scattering, such as flexible fiber-optic Raman probes that can be effectively used in clinical conditions and portable spectrometers [7,8]. However, despite some advances in technical equipment, the analysis of experimental Raman spectra still remains challenging. The reason for this is that Raman spectra contain an extremely large amount of information on all the chemical substances that can be found in skin. Many components of skin tissue that are accepted as biomarkers have similar peak positions in Raman spectra and overlap the spectrum of other substances such as the extracellular matrix, etc. Moreover, in addition to differences caused by the disease development, there are also some within-group differences associated with comorbidities and skin phenotypes. It is also important to understand that the quality of the Raman spectra recorded using portable Raman probes is usually slightly lower than the spectra recorded using a Raman microscope [4,7]. This can be explained by the fact that Raman microscopy is usually applied to investigate thin histological sections of tissues, while in the case of thick samples or in vivo studies, in addition to the useful Raman signal, we also reveal the contribution from the surrounding components, which cannot be removed by focusing. The spectra may also contain a signal associated with the optical system.

In this research, we use a portable Raman setup [7]. The methodology for the evaluation of the effectiveness of a portable setup and SNR estimation are described in [7].

Among the methods of Raman spectra analysis, principal component analysis (PCA) can be noted. It is an orthogonal linear transformation, which is commonly used for dimensionality reduction by projecting data onto the new principal components. The analysis of Raman spectra by the PCA was successfully applied in [9]. However, despite the positive experience, a significant limitation of this method in terms of its application to real data is non-physicality, i.e., the impossibility of the physical interpretation of the components, since they “absorb” the variance from several initial variables.

Another approach applied to the analysis of Raman spectra is neural networks. They are used as a classifier of diseases, as for example was undertaken in [10,11,12]. The main disadvantage is that the use of neural networks requires a lot of training and test datasets. For multivariate data discrimination, the method of discriminant analysis with the projection on latent structures (PLS-DA) is also widely used in chemometrics, biology, and medicine. It finds a linear regression model by projecting the predicted variables and the observable variables to a new space. In other words, the PLS-DA is able to construct a regression model based on the significant spectral differences [4,5,6,7,13]. However, these black-box approaches do not essentially analyze the spectra, focusing exclusively on the classification problem, and the classification criteria in many cases remain unknown.

To provide some clear, physically-based rules for discriminating Raman spectra, it is preferable to use white-box approaches when we know and understand how the algorithm of analysis works and in which physical or medical criteria are used to classify spectra. This approach is shown in [14], where a two-step phase-type method based on the analysis of spectral intensity alteration in the tumor tissue in comparison with the healthy tissue near the lesion is used for tumor detection and identification.

However, the white-box approach has limitations, due to the lack of information on the objects under study—in this case, skin tissue and tumors. We do not have complete data on the composition of the skin and the changes in composition that occur with the development of the disease. Using only well-known and understandable criteria to analyze Raman spectra, we risk losing classification accuracy as a result.

In this paper, we propose using a gray-box approach, when a model has a partial theoretical structure and some unknown parts derived from the data. To analyze the low signal-to-noise ratio (SNR) Raman spectra of skin, we propose using the multivariate curve resolution alternating least squares (MCR-ALS) method.

The MCR-ALS method is widely used to reconstruct concentration profiles in chemical analysis [15,16]. The main advantage of MCR analysis in a fully unsupervised mode with the unknown basis is its ability to physically interpret the components into which the original Raman spectra are divided. As a result of this analysis, we obtain the spectra of the substances constituting the sample and their contribution to the Raman spectrum. Their contribution, in fact, is directly related to the real concentrations of the substances.

On the other hand, the MCR-ALS analysis cannot be explicitly called a white box. Therefore, the research problem is to choose the optimal number of the components into which the spectra will be unmixed. It may depend on both the study objective and the characteristics of the Raman spectra. The MCR-ALS analysis combines the white-box and the black-box approaches and thus can be called a ‘gray-box’ approach.

Recently, the MCR-ALS method has found a wide application in biology and medicine to analyze spectral data, when it is necessary to determine the concentrations of components of a mixture from the spectra [16,17]. For example, Xu and Rice [18] used MCR-ALS spectral unmixing in fluorescence imaging. Chen et al. [19] used the Raman spectroscopic detection of keratin with the MCR-ALS analysis for automatic oral cancer diagnosis. Iwasaki et al. [20] investigated the possibilities of discriminating breast-cancer cells from normal mammary epithelial cells by Raman microspectroscopy and MCR-ALS analysis. In [21], samples of biological tissues were studied in situ using a Raman scanning microscope. Using the MCR-ALS method, the authors were able to obtain the spectra of Raman active skin components involved in biochemical changes during the development of various neoplasms. In [22,23], the researchers used MCR-ALS for the quantitative analysis of pharmaceutical and agricultural samples, as well as for vibrational spectroscopic image analysis of biological material.

This work is devoted to the study of the possibilities of the MCR-ALS method for analyzing skin tissues in vivo. In a clinical setting, high-quality recording of Raman spectra can be difficult due to a large patient flow and the limited time available for patient examination. An SNR is determined by multiple factors, including the laser power, the detector cooling temperature, and the signal-accumulation time. It was revealed that there is a linear relationship between SNR and temperature (see [24]). Although it is possible to increase the SNR due to the strong cooling of the detector, the portability will be lost. Raman microscopy is usually used to investigate thin sections of tissues, but if we investigate tissues in vivo, it is expensive and, in most cases, inefficient to use Raman microscopy. Therefore, it can be assumed that the skin Raman spectra subjected to analysis by the MCR-ALS method will have a rather low SNR. 

For example, in [4,5,6,7], a portable spectroscopic device able to record Raman spectra, with a signal-to-noise ratio equal to three, is used to record the Raman spectra of the skin. We take into account the fact that the Raman spectra of some of skin-tissue components also have similar peaks and bands, for example, collagen, elastin, and keratin [21,25,26,27]. Nevertheless, we believe that it is possible to distinguish groups of components, for example, proteins, the moisturizing factor, and pigments (melanin), since some of the Raman peaks of the selected groups are located in different spectral bands. Our results show the MCR-ALS analysis of experimental in vivo Raman spectra of normal skin and tumors as well as the spectra which correspond to the contribution of the optical system and skin components. In addition, we investigated the possibilities of using the spectral profiles obtained by MCR-ALS analysis to classify skin tissues.

## 2. Materials and Methods

### 2.1. Experimental Data

The Raman spectra were recorded using a portable spectroscopic setup, which includes a thermally stabilized LML-785.0RB-04 diode laser module as an excitation source (785 ± 0.1 nm central wavelength) and a QE 65 Pro spectrometer (OceanOptics, Inc., Dunedin, FL, USA) with a CCD detector operating at −15 °C [4]. Spectra registration was carried out with a spectral resolution of 0.2 nm in the range from 837 to 920 nm, which corresponds to 792–1874 cm^−1^.

All in vivo spectra measurements were performed with a signal-accumulation time of 60 s. The laser power density on the skin was about 0.3 W/cm^2^, which does not exceed the ANSI maximum permissible exposure limit of 1.63 W/cm^2^ and does not cause any damage to the skin or discomfort to patients.

In this case, the SNR of the recorded Raman spectra is 3 in accordance with the methodology for SNR estimation described in [7].

In Section 3.1, we used the recorded raw spectra, and in Section 3.2 and Section 3.3, the spectra were subjected to preprocessing in the following way: the spectra were cut in the range from 860 to 920 nm, which corresponds to 1114–1874 cm^−1^. The spectra were then preprocessed with baseline removal with asymmetric least squares (lambda = 6, *p* = 0.1, number of iterations = 10) and smoothing by the Savitzky–Golay method (derivative order = 0, window width = 15, polynomial order = 1) using the baseline package of the RStudio software [28]. An example of the experimental Raman spectra of normal skin before and after preprocessing is shown in Figure 1. It is clearly seen that before preprocessing, in addition to the useful Raman signal, the spectra also contain a fluorescence signal.

It should be noted that preprocessing methods can probably influence the results of the MCR-ALS. A particularly high influence is observed for cases with a fluorescence background and without fluorescence (see Section 3.1 and Section 3.2). In our case, preprocessing increases the number of reconstructed components (see Figure 2). The issue of the influence of preprocessing methods on the results of MCR-ALS analysis of Raman spectra of the skin is subject to separate research.

The in vivo study of skin tumors was performed at Samara Regional Clinical Oncology Dispensary. The study involved 540 patients. The Raman spectra of normal skin and skin neoplasm were recorded for each patient. In total, we used 1000 spectra: 540 of normal skin (NS), 113 of keratosis (K), 122 of basal cell carcinoma (BCC), 67 of malignant melanoma (MM), and 158 of pigmented nevus (PN). All in vivo studies were conducted on patients over 18 years of age. Informed consents were acquired from all the patients before the in vivo study. The studies were approved by the Ethics Committee of Samara State Medical University (Samara, Russia).

In addition to the Raman spectra, we have obtained information on the actual diagnoses for each patient. These diagnoses were made by the doctors of Samara Regional Clinical Oncology Dispensary using histological examination of tumor samples.

### 2.2. Multivariate Curve Resolution Alternating Least Squares Analysis

In this study, we use the multivariate curve resolution alternating least squares (MCR-ALS) method to analyze the Raman spectra of skin tissue and distinguish the disease markers. MCR is a group of methods which can be used to solve the curve resolution problem in spectroscopy, and ALS is an algorithm that is used for the MCR basic bilinear model.

The MCR problem can be defined as follows. Suppose we have a mixture of A components. Every individual component is usually called a ‘pure component’. Every pure component has a spectrum, which can be represented as a column vector s_i_ with size J, which is a number of values in each spectrum (corresponding to the number of wavelengths or wavenumbers).

If we mix the components into one mixture and take a spectrum of this mixture, the spectrum will be just a linear combination of the spectra of the pure components [23]. This can be written as follows:d = c_1_s^T^_1_ + c_2_s^T^_2_ + ⋯ + c_A_s^T^_A_,(1)
where d is a vector of spectral values representing the spectrum of the mixture (J); c_1_, c_2_, …, c_A_ are concentrations of the pure components in the mixture; and s_1_, s_2_, …, s_A_ are the spectra of the pure components.

The equation can be written in a more compact form:D = CS^T^,(2)
where C is a matrix of concentrations of the pure components in the mixture, and S is a matrix of the spectra of the pure components.

The task of the MCR methods is to get C and S by knowing D. This is not a trivial task, as the expression above does not have a unique solution. For example, one of the solutions is what the PCA gives; however, the scores do not correspond to the real concentration values nor do the loadings represent the spectra of the pure components. In fact, it is impossible to obtain C and S precisely; what we obtain is a sort of estimate, and we can rewrite the equation as:D = C∙S^T^ + E,(3)
where E is the error matrix [23].

The main advantage of MCR analysis is the ability to physically interpret the components into which the original Raman spectra are divided. As a result of the analysis, we obtain the spectra of the substances constituting the sample and their contribution to the Raman spectrum. Its contribution, in fact, is directly related to the real concentrations of the substances. Among the limitations of the MCR method is the difficulty of unmixing substances with similar Raman spectra, when the substance does not contain unique Raman peaks in the spectrum [23]. In addition, the research problem is to choose the optimal number of components into which the spectra will be unmixed. It may depend on both the study objective and the characteristics of the Raman spectra.

It should be specially noted that MCR analysis could be sensitive to noise in the analyzed Raman spectra. In practical application, the efficiency of evaluating the component concentration in a test sample can decrease due to the fact that the spectra contain a noise signal caused by the optical system or the fluorescence signal.

Previously, we studied the possibility of applying the MCR-ALS method to the analysis of Raman spectra of proteinogenic amino acid mixtures [29] and the effect of noise in Raman spectra on the analysis result [30]. Our studies demonstrate the possibility of unmixing several spectrally similar components of one complex mixture under noisy conditions. Using the MCR-ALS method makes it possible to successfully estimate the amino acid concentrations in a mixture, even when the Raman spectra have a low SNR (from 3 to 5).

For unmixing the spectra, we used the MCR-ALS GUI v4c by András Gorzsás [31]. To analyze experimental skin Raman spectra, we utilized the MCR-ALS method, and in this case we do not know the exact number of substances contained in the skin sample. Feng et al. [21] report that eight components captured the skin constituents as measured in in vivo human skin cancers. However, this also depends on the characteristics of the Raman spectra. Therefore, it should be investigated how many components it is necessary to obtain in the original Raman spectra for an optimal result. 

This can be undertaken based on a priori knowledge or by analyzing the dataset using singular value decomposition (SVD) [31]. Figure 2 shows the numerical results of the singular value decomposition for set of raw Raman spectra and set of preprocessed Raman spectra. The number of components is usually selected, after which a large drop in eigenvalues is observed or the eigenvalues approximate 0. In both cases, Figure 2 displays a gradual decrease with no abrupt drop in eigenvalues. Thus, at first glance, the number of components for set of raw Raman spectra can reasonably be assumed to be 2 (3 is already close to 0 on the SVD plot). The number of components for a set of preprocessed spectra can reasonably be assumed to be 5–6 (7 is already close to 0 on the SVD plot). That is, in our case, preprocessing increases the number of reconstructed components (see Figure 2). However, it is further shown that the choice of the number of components according to the SVD results does not allow the MCR analysis to reconstruct diagnostically useful spectral profiles (see Section 3.1). Therefore, we chose the number of components based on a priori knowledge and other considerations.

The estimation of the initial values was performed using a SIMPLISMA-based algorithm with a default value of 10% allowed noise [32].

As constraints, we used a default spectrum equal-length closure constraint (this is the most common closure constraint for general spectroscopic data and means an equal spectral-area normalization for the MCR-ALS algorithm to avoid possible scale indeterminacies) and a non-negativity constraint in both the spectrum and the concentration profiles, which is also hard-coded in the GUI with no user control. It is implemented as a mild constraint, using a fast non-negative least squares algorithm [31]. 

In the case of using the spectrum basis in Section 3.3 (which is described below), we also used equality constraints for spectra profiles. Equality constraints allow the user to fix values in the spectral profiles, such as the spectrum of a known compound present in the dataset [31].

The maximum number of iterations that will be performed, if convergence or divergence does not happen before, was set to 150. The convergence criterion (the percentage of change in the standard deviation of residuals between two consecutive iterations) was set to 0.1 (default setting). In other words, the convergence is achieved when the change in the standard deviation of the residuals is lower or equal to 0.1% between two consecutive iterations. 

After selecting the number of components, setting the constraints and the end conditions for the modelling, the MCR-ALS algorithm can be started. After initial estimation is given for C, it is optimized iteratively using the ALS algorithm until the convergence condition is met and the maximum number of iterations is reached [31].

In Section 3.1, we use 250 in vivo Raman spectra: 100 of normal skin, 90 of BCC, and 60 of MM. The Raman spectra were not preprocessed, which means that the signal associated with fluorescence was retained. The spectra were decomposed by the MCR-ALS method without using a basis. The number of the decomposed spectra varied, and for visual analysis of the obtained spectra, the number of the decomposed spectra was chosen to be nine as an optimal solution.

In Section 3.2, the preprocessed Raman spectra were studied: 540 spectra of normal skin, 113 of K, 122 of BCC, 67 of MM, and 158 of PN. The spectra of normal skin were divided into a different number of components from 4 to 30, and the spectra of the diseases into 5 components.

In Section 3.3, we also investigate 540 spectra of normal skin, 113 of K, 122 of BCC, 67 of MM, and 158 of PN. However, there is one important difference between Section 3.2 and Section 3.3. In Section 3.2, only matrix D is known, while matrices C and S^T^ remain undefined. In Section 3.3 we also define matrix S^T^.

In other words, in Section 3.2. we only know the matrix of the original Raman spectra, that is, the matrix of mixtures, and the matrices of concentrations of the components and the spectra of the components are not known. In Section 3.3, in addition to the matrix of mixtures, we also know the matrix of the spectra of the components; only the matrix of the concentrations of these components remains unknown. In this case, the matrix of the spectra of the components is called a ‘basis’ or ‘basic spectra’.

The fact is that in the case of MCR-ALS analysis, without using a basis we cannot unambiguously determine into which specific components the Raman spectra are divided (for example, to which specific substance each selected component corresponds). Using a basis, we can solve this problem. In this case, the set of components into which the Raman spectra are divided remains fixed and uniquely determined. As a result, we can unambiguously determine what the concentrations of the components in the tissue sample are. The disadvantage of this approach is the fact that we often do not know what components the tissue sample consists of, so it is not possible to set the basis in advance.

We solve this problem in the following way. First, we subject the Raman spectra to MCR-ALS analysis without using a basis (see Section 3.2). In Section 3.3, as a basis, we use several spectra obtained in Section 3.2, that, in our opinion, most accurately represent the contribution of some skin components. In addition to the fixed bases, we set the possibility of using four additional bases, which are identified in advance. In other words, the Raman spectra were unmixed into eight components, four of which were strictly defined. 

Spectra equality is used to fix spectral profiles. For all the components that are to be constrained, the row contains numbers (i.e., spectral intensities at each wavenumber); for unconstrained components, the row contains NaN values (see [31]).

### 2.3. Analysis of Results

As a result of the MCR-ALS method, two matrices were obtained: a matrix of the component concentrations and a matrix of their Raman spectra. It is important to understand that the values of the concentrations and intensities of the spectra were obtained in arbitrary units and vary for each individual analysis. The reason for this is the ambiguity of the bilinear transformation, as well as the fact that the raw Raman spectra are also recorded in arbitrary units. Thus, the Raman spectra were normalized with rescaling (min-max normalization, when the lowest intensity is set to 0, and the highest is set to 1) and analyzed for the presence of Raman bands and peaks as well as their ratio.

In Section 3.3, the results were presented using a box-plot diagrams with a spread of the concentrations and the receiver operating characteristic (ROC) curves AUC with 95% confidence interval (CI). The box-plot diagrams and ROC curves were plotted using the pROC package of the RStudio software [28].

A box plot shows the distribution of quantitative data in a way that facilitates comparisons between variables or across levels of a categorical variable. The box shows the quartiles of the dataset while the whiskers extend to show the rest of the distribution, except for points that are determined to be outliers.

An ROC curve illustrates the diagnostic ability of a binary classifier system as its discrimination threshold is varied. The ROC curve is created by plotting the true positive rate (TPR) against the false positive rate (FPR) at various threshold settings. The true-positive rate is also known as sensitivity. The false-positive rate is also known as probability and can be calculated as (1 − specificity).

The area under the ROC curve (AUC) provides an aggregate value of how well the model correctly classifies the cases with all possible cutoff values. AUC values range between 0.5 and 1, where an area of 0.5 means that the model predicts which outcomes will be 1 or 0 no better than flipping a coin, and an area of 1 means that the model predicts perfectly.

Following this, the profiles of the obtained component concentrations were subjected to logistic regression. Logistic regression is a form of multiple regression whose general purpose is to analyze the relationship between the multiple independent variables (called predictors) and the dependent variable [33]. In our case, the predictors are the concentration profiles of the components representing their contribution to the Raman spectrum. We applied binary logistic regression for two cases:malignant neoplasms versus benign neoplasms;malignant melanoma (MM) versus pigmented neoplasms (K + PN).

This type of statistical model is often used for classification in medicine and biology. Binary logistic regression estimates the probability of an event occurring, such as melanoma or no melanoma, based on a given dataset of independent variables. In other words, the goal of binary logistic regression is to simply classify observations as one of the two possible outcomes that the model describes. 

That is, for these two cases, we obtained the probabilities of the presence of a particular disease in each patient. Besides this, at the first step of logistic regression, the significance of all predictors was calculated, which represents their influence on the regression result, that is, on the predicted value. These probabilities were called predicted values. Since the outcome is a probability, the dependent variable is bounded between 0 and 1. Logistic regression was performed using the SPSS Statistics software [33].

In addition, the profiles of the component concentrations were subjected to partial least square discriminant analysis (PLS-DA) using the pROC package of the RStudio software [27]. 

Partial least squares discriminant analysis (PLS-DA) is one of the most widely used classification techniques in many fields, and generates a large amount of data, such as in spectroscopy. PLS-DA is a supervised clustering or classification method, used to optimize separation between different groups of samples, which is completed by linking two data matrices: X (i.e., raw data) and Y (i.e., classes or diagnoses). This method aims to maximize the covariance between the independent variables X and the corresponding dependent variable Y of highly multidimensional data by finding a linear subspace of the explanatory variables. The new subspace permits the prediction of the Y variable based on a reduced number of factors (PLS components).

To check the classification models stability, a 10-fold cross-validation was performed. The predicted values were presented using box-plot diagrams with a spread of the concentrations and the ROC curves. The box-plot diagrams and the ROC curves were plotted using the pROC package of RStudio software [28].

## 3. Results and Discussion

### 3.1. Unmixing the In Vivo Raman Spectra without Processing

As described in Section 2.2, we conducted the MCR-ALS analysis of 250 raw in vivo Raman spectra: 100 of normal skin, 90 of BCC, and 60 of MM. The spectra were decomposed without using a basis. The number of the decomposed spectra, i.e., the spectra of pure components, varied, and using the visual analysis of the obtained spectra, the number of pure components was chosen to be nine as an optimal solution. The spectra of these components are shown in Figure 3.

As can be seen in Figure 3, most of the unmixed components correspond to the fluorescence signal (see Components 2, 3, 4, 5, 6). This is fully consistent with the known information on the significant contribution of fluorescence to the total signal and indicates the need to remove the fluorescence signal from the spectrum if we need to study the Raman signal. Moreover, in most of the spectra obtained (see Components 3, 4, 5, 6, 7, 9), a sinusoidal component is visible. We believe that this component is related to the contribution of the optical system. The reason may be optical filters in the setup for recording Raman spectra. 

As an example of such a sinusoidal component, Figure 4 shows the spectrum of the 785 nm RazorEdge^®^ ultrasteep long-pass edge filter by Semrock [34]. A component representing the contribution of optics (but without a sinusoidal pattern) was obtained in [35], where the authors applied a non-negative matrix factorization (NNMF) algorithm to the in vivo Raman spectra of the upper epidermis. It can be assumed that the source for the sinusoidal pattern is an etaloning effect from the back-thinned CCD of the spectrometer. The reason for this sinusoidal effect and ways to reduce it will be the focus of our further research.

Section 3.2 gives evidence that the preprocessing we use is not able to filter out this component completely.

In addition, in Figure 3, Raman peaks are seen in the spectra of Components 7 and 8. The peaks at 1316 and 1660 cm^−1^ are included in the bands of amide III and amide I and correspond to proteins [25]. If we talk about the components of the skin, then these peaks can correspond to collagen, elastin, and keratin [21]. The peak at 1450 cm^−1^ is associated with bending CH_2_/CH_3_, scissoring CH_2_/CH_3_, and bending CH_2_. This peak may correspond to the lipids that protect the skin from environmental factors. Those may be triolein or ceramide [21]. Proteins and lipids are the main constituents of the upper layers of skin tissue, so it is not surprising that they contribute so much to the skin spectrum and can be distinguished by MCR-ALS even in the presence of fluorescence.

### 3.2. Unmixing Preprocessed In Vivo Raman Spectra

As described in Section 2, we preprocessed the in vivo Raman spectra of normal skin and diseases. Further, as discussed in Section 2.2, we conducted the MCR-ALS analysis of 1000 preprocessed in vivo Raman spectra: 540 of normal skin, 113 of K, 122 of BCC, 67 of MM, and 158 of PN. The spectra of normal skin were divided into a different number of components from 4 to 30, and the disease spectra into 5 components. All the component spectra obtained are presented in the Appendix A.

Spectra analysis revealed that with a large number of components into which the Raman spectra are unmixed, MCR-ALS reconstructs single peaks, and with a decrease in the components, these peaks are combined into more complex spectra. Below we describe only some of the spectra, which, in our opinion, are most similar to the Raman spectra of the components that make up the skin tissue (Figure 5A–D).

Component 1 (Figure 5A) corresponds to the contribution of the optical system. A similar component representing the contribution of optic filter is shown in Figure 4. The contribution of microscope optics in Raman spectra was highlighted in [35], where the authors applied an NNMF algorithm to the in vivo Raman spectra of upper epidermis. It seems quite logical that in [35] this component was one of the last obtained, and in our study, the first. The reason for this is that we used a spectroscopic setup with a low signal-to-noise ratio instead of a Raman microscope.

The ability to identify the parasitic signal associated with the contribution of the optical system can potentially be used in preprocessing the recorded Raman signals. In the spectrum of Component 2 (Figure 5B), the peaks at 1174 and 1230 cm^−1^ correspond to the natural moisturizing factor of the skin [35]. The peaks at 1390 and 1520 cm^−1^ are the linear stretching of the C-C bonds within the rings and the in-plane stretching of the aromatic rings of melanin [21,35,36]. The area from 1650 to 1800 cm^−1^ is due to C=C and C=O stretching and corresponds to skin lipids. In general, the most intense regions in these spectra are the melanin regions, so the second component can be considered predominantly melanin. A similar shape of the spectrum was obtained in [35], where melanin exhibits broad Raman bands in the 1200–1700 cm^−1^ range with two local maxima at ≈1370 and 1580 cm^−1^.

In the spectrum of Component 3 (Figure 5C), one can see four peaks. The peak at 1150 cm^−1^ corresponds to the contribution of the natural moisturizing factor, and that at 1280 cm^−1^ to amide III and urocanic acid [35,37]. The most pronounced peak at 1450 cm^−1^ corresponds to the C–H bending proteins and may be contributed by keratin, collagen, and elastin [21]. In addition, the peak at 1450 cm^−1^ may also correspond to the contribution of urocanic acid [35]. The peak at 1660 cm^−1^ is the amide I of proteins and may be contributed by collagen [25,38]. In other words, Component 3 represents the contribution of various proteins of the skin tissue and, possibly, of the natural moisturizing factor, including urocanic acid. Moreover, the contribution of collagen is so high that the peaks at 1450 and 1660 cm^−1^ are visible even when the raw Raman spectra of skin with fluorescence are decomposed (see Section 3.1). Comparison with the results obtained by Feng et al. shows the similarity of this spectrum with the spectra of collagen, elastin, and keratin in the range from 1400 to 1800 cm^−1^. Thus, in [21] we can also see the peaks corresponding to amide III (1269 cm^−1^), to C–H bending proteins (1450 cm^−1^), and C–O amide I vibration (1665 cm^−1^). Nevertheless, Feng et al. were able to distinguish the spectra of collagen, elastin, and keratin separately, while we obtained only the spectrum of their mixture.

In the spectrum of Component 4 (Figure 5D), the most intense peak is at 1650 cm^−1^ and corresponds to water. Comparison with the results by Feng et al. [21] shows that the spectra match almost completely.

The other components reconstructed as a result of the MCR-ALS analysis (see the Appendix A) apparently correspond to the contribution not of a specific skin component or related substances, but of a group of components. This may be attributed to superposition of peaks and bands, which prevents the MCR-ALS method from working correctly.

It is important to understand that the reason for the superposition of peaks and bands is not only the nature of these substances. The quality of the equipment used also plays a great role. The overlapping of spectral peaks and bands occurs in the presence of intense noise, i.e., in the Raman spectra with a low signal-to-noise ratio.

Feng et al. used Raman microscopy to record the spectra and identified eight different components [21]. Yakimov et al. also used Raman microscopy but applied an NNMF algorithm instead of MCR-ALS and identified ten different components [35]. In our case, using a portable conventional Raman spectroscopy setup, we managed to explicitly reconstruct four components, whereas the rest of the components represent the contribution of several completely different groups of substances. Nevertheless, we can use the four spectra of components obtained as a basis and evaluate their concentration in the skin Raman spectra.

In Section 3.3, we also investigate 540 spectra of normal skin, 113 of K, 122 of BCC, 67 of MM, and 158 of PN. However, there is one important difference between Section 3.2 and Section 3.3. In Section 3.2, only matrix D is known, while matrices C and S^T^ remain undefined, whereas in Section 3.3 we also define matrix S^T^.

In other words, in Section 3.2. we only know the matrix of the original Raman spectra, that is, the matrix of mixtures, while the matrices of concentrations of the components and the spectra of the components remain unknown. In Section 3.3, in addition to the matrix of mixtures, we also know the matrix of the spectra of the components; only the matrix of the concentrations of these components remains unknown. In this case, the matrix of the spectra of the components is called a basis or basic spectra.

### 3.3. The Contribution of Individual Components to the Raman Spectra of Various Diseases

As described in Section 2.2, we also investigated 1000 preprocessed in vivo Raman spectra: 540 of normal skin, 113 of K, 122 of BCC, 67 of MM, and 158 of PN. The only difference is that we used the MCR-ALS method with a basis that contained four fixed spectra (see Figure 5A–D). MCR-ALS is performed with 99.4893% explained variation and 6.711% lack of fit (these indicators are detailed in [31]). As a result, we obtained four additional Raman spectra. These spectra are presented in Figure 4E–H. The concentrations of the components were rendered using box-plot diagrams. They are presented in the Appendix A.

As one can see in Appendix A, Components 1, 3, and 4 (proteins, water, and optical contribution; see Figure 5A,C,D) are expressed almost identically in the spectra of all diseases. The most striking differences are seen in the example of Component 2 (Figure 5B). As we mentioned earlier, Component 2 presumably represents the contribution of melanin. The concentrations of Component 2 are high in K, MM, and PN, and low in normal skin and BCC. It is well-known that, on average, the content of melanin in K, MM, and PN is higher than in normal skin [21].

Among the box-plots of additional components (Figure 5E–H), the box-plots of Components 5 and 7 (Figure 5E,G) should be distinguished. As can be seen in Appendix A, Component 5 is less pronounced in K and BCC and is more pronounced in PN and MM. Referring to the spectrum of this component (Figure 5E), a band from 1150 to 1320 cm^−1^ of amide III can be distinguished. This spectrum may correspond to the contribution of lipids and natural moisturizing factor [24,35,38]. Besides this, in the spectra of Components 5 and 6 (Figure 5E,F) we see an intense peak at 1240 cm^−1^ that corresponds to the contribution of proteins. If we compare Components 3, 5, and 6 with the spectra of collagen, elastin, and keratin, it can be noted that Components 5 and 6 somehow complement the spectrum of Component 3 in the range from 1150 to 1320 cm^−1^. The contribution of proteins might have been distributed among these components. Component 7 (Figure 5G) is most pronounced only in BCC, and its contribution is extremely insignificant in the Raman spectra of other diseases. We can note peaks at 1144 cm^−1^ (natural moisturizing factor); 1275 and 1750 cm^−1^ (lipids); 1355, 1386, and 1559 cm^−1^ (melanin); and 1694 cm^−1^ (collagen) [21,35]. In the spectrum of Component 8 (Figure 5H) one can see intense peaks at 1450 and 1650 cm^−1^, which suggests that this is also part of the total contribution of proteins and water. The spectrum of Component 6 (Figure 5F) shows an intense band at 1700–1850 cm^−1^, which may correspond to the contribution of lipids [21].

Thus, Components 2–4 represent the contribution of certain substances or groups of substances (e.g., proteins), whereas Components 5–8 represent the contribution of several completely different groups of substances with a set of overlapping peaks, which leads to their ratio redistribution in different diseases.

The limitation of this study lies in the fact that we cannot compare the concentration values of skin components (i.e., the values of the contribution of the reconstructed spectra to the total Raman spectrum) with their true values. We can only indirectly confirm general patterns, noting, for example, an increased average level of melanin in pigmented neoplasms. 

Among all box-plot diagrams in Appendix A, we selected several most successful cases, where we can note the possibility of classifying the Raman spectra by diseases. Figure 6 and Figure 7 show the box-plot diagrams and ROC curves corresponding to those cases. 

More advanced discrimination models demonstrate higher ROC AUC values; therefore, it seems plausible to compare the ROC AUCs of the constructed classification models. In the case of BCC (n = 122) vs. normal skin and all diseases (n = 878) neoplasm classification, the highest ROC AUC is 0.772 (0.720–0.823, 95% CI) when discriminated by Component 3 and a slightly lower ROC AUC is 0.685 (0.634–0.736, 95% CI) when discriminated by Component 7. The best separation of BCC (n = 122) vs. MM (n = 67) is possible by Components 5 and 7 demonstrating 0.700 (0.626–0.773, 95% CI) and 0.690 (0.623–0.757, 95% CI) ROC AUCs, respectively. In the case of malignant (MM, BCC; n = 189) vs. benign (K, PN; n = 271) neoplasm classification, the highest ROC AUC is 0.653 (0.602–0.705, 95% CI) when discriminated by Component 3 and a lower ROC AUC is 0.614 (0.570–0.658, 95% CI) when discriminated by Component 7. One can notice that we have better discrimination models using components with a lower number (e.g., Components 3 or 5), that is, the components that are reconstructed first by the MCR-ALS method.

Interestingly, the discrimination models of MM (n = 67) vs. PN (n = 158) demonstrate 0.656 (0.574–0.738, 95% CI) ROC AUC. This is a fairly high result, given the fact that these types of neoplasms have similar external manifestations and a comparable amount of melanin [29,35].

We suppose that using more accurate equipment to record Raman spectra (e.g., a Raman microscope) might lead to better results. However, the use of microscopes makes the in vivo research unnecessarily complicated. Partial application of Raman microscopy could be a compromise solution. For instance, we can record Raman spectra of relatively homogeneous samples in terms of component composition on a microscope (as was performed in [21]) and use these spectra as a basis. Moreover, the Raman spectra to be analyzed should be further recorded with a Raman probe. This approach can improve the quality of spectrum analysis. Returning to the use of microscopy, MCR-ALS can be applied to the analysis of the Raman spectra recorded with a microscope, for example, for the analysis of biological fluids or biopsy material.

Furthermore, taking into account the obtained concentrations (see Appendix A), as described in Section 2.3, we applied the binary logistic regression for two cases:malignant neoplasms versus benign neoplasms;malignant melanoma (MM) versus pigmented neoplasms (K + PN).

Using the results of the binary logistic regression we obtained box-plot diagrams and ROC-curves of the predicted values (see Figure 8).

In the case of malignant (MM, BCC; n = 189) vs. benign (K, PN; n = 271) neoplasm classification, the ROC AUC is 0.698 (0.650–0.746, 95% CI), and in the case of MM (n = 67) vs. pigmented neoplasms (n = 271) classification, the ROC AUC is 0.702 (0.629–0.776, 95% CI). 

In [4], the data were processed by means of regression analysis using PLS-DA, and the AUCs of similar classification models were 0.75 (0.71–0.79) and 0.61 (0.53–0.69), respectively. Thus, we can conclude that PLS-DA (applied to Raman spectra, as is described in [4]) and MCR-ALS analysis followed by binary logistic regression produce nearly the same results.

The significance values of all predictors (concentrations of the components) are presented in Table 1.

As can be seen from Table 1, in the case of the classification of malignant vs. benign tumors, significant components are 1, 3, 4, 7, and 8. Earlier, we assumed that these components correspond to optical system contribution, proteins, water, lipids, nature moisturizing factor, and melanin. Feng et al. [21] also report about the changes in the concentration of collagen during disease development; for instance, in the case of BCC, the relative concentration of collagen decreases, while in the case of PN, it increases. It should be noted that Component 2 is expected to become less significant due to the presence of melanin in both groups.

In the case of the classification of MM vs. pigmented neoplasms (K + PN), Component 3, which reflects the contribution of proteins, becomes less important. Indeed, Feng et al. [21] reports that with the development of K and MM, the share of collagen changes in approximately the same way. Components 6, 7, and 8 are also become less significant, because they also represent the contribution of proteins. 

Among the box-plots of additional components (Figure 5E–H), the box-plots of Components 5 and 7 (Figure 5E,G) should be distinguished. As can be seen in Appendix A, Component 5 is less pronounced in K and BCC and is more pronounced in PN and MM. Referring to the spectrum of this component (Figure 5E), the band from 1150 to 1320 cm^−1^ of amide III can be distinguished. This spectrum may correspond to the contribution of lipids and natural moisturizing factor [25,35,38]. Besides this, in the spectra of components 5 and 6 (Figure 5E,F) we see an intense peak at 1240 cm^−1^ that corresponds to the contribution of proteins. If we compare Components 3, 5, and 6 with the spectra of collagen, elastin, and keratin, it can be noted that Components 5 and 6 somehow complement the spectrum of Component 3 in the range from 1150 to 1320 cm^−1^. It is possible that the contribution of proteins was distributed among these components. Component 7 (Figure 5G) is most pronounced only in BCC, and its contribution is extremely insignificant in the Raman spectra of other diseases. We can note peaks at 1144 cm^−1^ (natural moisturizing factor); 1275 and 1750 cm^−1^ (lipids); 1355, 1386, and 1559 cm^−1^ (melanin); and 1694 cm^−1^ (collagen) [21,35]. In the spectrum of Component 8 (Figure 5H) one can see intense peaks at 1450 and 1650 cm^−1^, which suggests that this is also part of the total contribution of proteins and water. The spectrum of Component 6 (Figure 5F) shows an intense band at 1700–1850 cm^−1^, which may correspond to the contribution of lipids [21].

It should be noted that the contribution of Components 1 (optical system contribution) and 7 (proteins, lipids, nature moisturizing factor, melanin) is small for both cases of classification. This is quite logical, because the contribution of the optical system is approximately the same for all recorded Raman spectra, and no significant changes in water during the development of diseases have been reported [21]. 

As described in Section 2.3, we applied the PLS-DA for two cases:malignant neoplasms versus benign neoplasms;malignant melanoma (MM) versus pigmented neoplasms (K + PN).

The box-plot diagrams and ROC-curves of the predicted values are presented in Figure 9.

In the case of malignant (MM, BCC; n = 189) vs. benign (K, PN; n = 271) neoplasm classification, the ROC AUC is 0.632 (0.581–0.683, 95% CI), whereas in the case of MM (n = 67) vs. pigmented neoplasms (n = 271) classification, the ROC AUC is 0.658 (0.583–0.733, 95% CI). Thus, we can conclude that PLS-DA applied to the profiles of the component concentrations gives approximately the same result as the logistic regression (cf. Figure 8 and Figure 9). 

## 4. Conclusions

The results obtained in this study demonstrate the possibility of unmixing several spectrally similar components using MCR-ALS analysis even under noisy conditions of the recorded Raman spectra. This gives grounds to talk about a possible effective use of the MCR-ALS method in portable devices for diagnosing neoplasms, when the speed and ease of use of the device are more important than its high spectral resolution.

Moreover, in addition to our initial problem of analyzing the Raman spectra in order to classify them, we were also able to use the MCR-ALS method to identify the parasitic signal associated with the contribution of the optical system. In other words, the method can be used as part of the Raman setup software to correct the setup bias. Eliminating the influence of optical equipment can be extremely useful in multicenter and transnational research, when completely different equipment with different technical characteristics is used to record Raman spectra. 

Unmixing such spectrally similar components as different proteins or lipids is still a challenge. In this study, we were unable to unmix them into separate components. The use of more localized measurement equipment for recording Raman spectra, for example, a Raman microscope, could make it possible to achieve better results. 

Besides this, one advantage of using Raman microscope is that in this case we can yield 2D microscopic imaging (see [21]) and build a direct relation between the skin constituents and the unmixed components by comparing them with the histology. 

In addition, the method of finding initial estimates (SIMPLISMA) suffers from an essential drawback, which consists of the need for “pure” variables [32], which cannot be expected because of the strong signal overlapping. Future studies should try other methods of finding initial estimates, for example, independent component analysis (ICA).

Another limitation of this study is the fact that we do not have an approved and confirmed method for determining the composition of the skin in vivo. We can only evaluate the MCR-ALS method using the classification models.

Nevertheless, despite some limitations of the method, we have obtained a satisfactory classification result for some diseases. We plan to devote our future research to supplementing our methodology with other methods for analyzing Raman spectra, such as using neural networks.

## Figures and Tables

**Figure 1 sensors-22-09588-f001:**
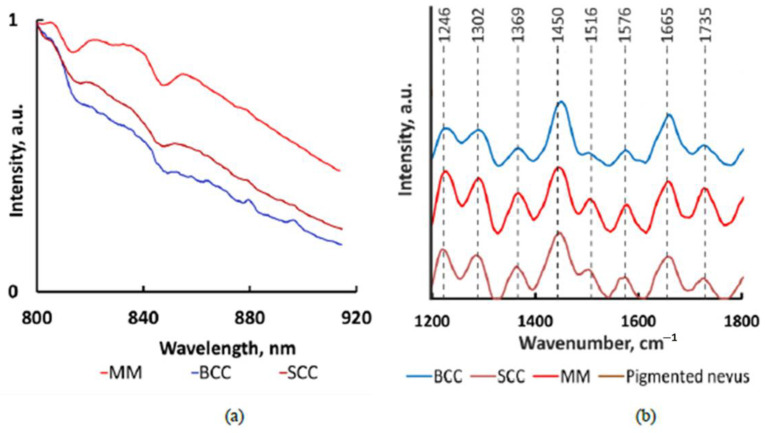
An example of in vivo Raman spectra before (**a**) and after (**b**) preprocessing: MM is malignant melanoma, BCC is basal cell carcinoma, SCC is squamous cell carcinoma. Adapted with permission from Ref. [13]. 2020, Y. Khristoforova.

**Figure 2 sensors-22-09588-f002:**
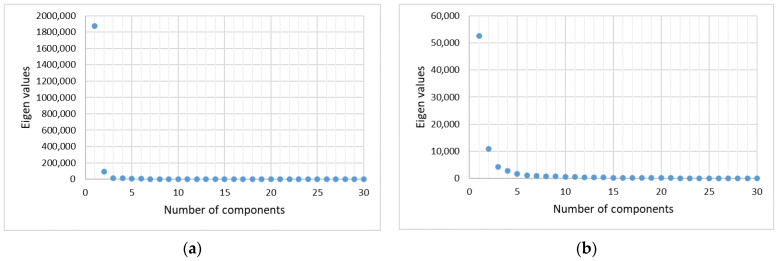
Determining the number of components by singular value decomposition: (**a**) set of raw Raman spectra, (**b**) set of preprocessed Raman spectra.

**Figure 3 sensors-22-09588-f003:**
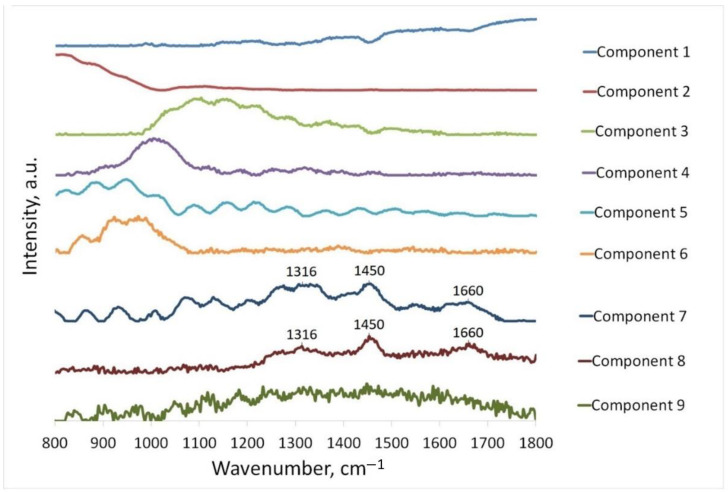
The spectra of skin components (presented in the order they were extracted as a result of MCR-ALS analysis).

**Figure 4 sensors-22-09588-f004:**
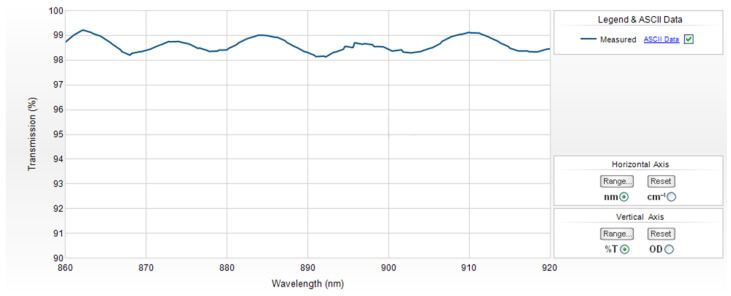
785 nm EdgeBasic™ best-value long-pass edge filter spectrum. Reprinted with permission from Ref. [34]. 2022, Semrock Optical Filters. © IDEX Health & Science, Semrock Optical Filters www.semrock.com (accessed on 11 August 2022).

**Figure 5 sensors-22-09588-f005:**
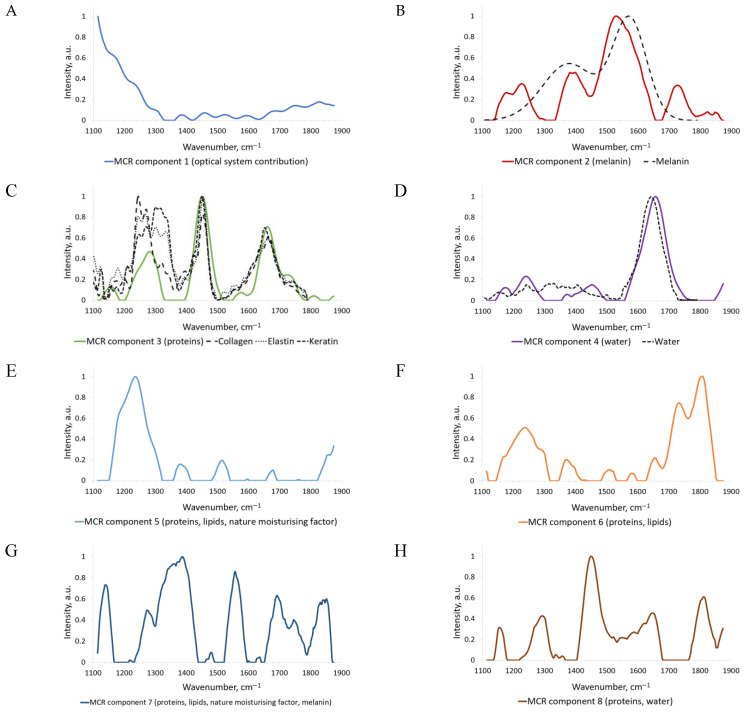
Raman spectra obtained by MCR-ALS analysis of the in vivo Raman spectra of skin (solid lines denote the spectra obtained in our study, dashed lines, the spectra obtained in [21]): (**A**) Component 1 (optical system contribution), (**B**) Component 2 (melanin), (**C**) Component 3 (proteins), (**D**) Component 4 (water), (**E**) Component 5 (proteins, lipids, nature moisturizing factor), (**F**) Component 6 (proteins, lipids), (**G**) Component 7 (proteins, lipids, nature moisturizing factor, melanin), (**H**) Component 8 (proteins, water).

**Figure 6 sensors-22-09588-f006:**
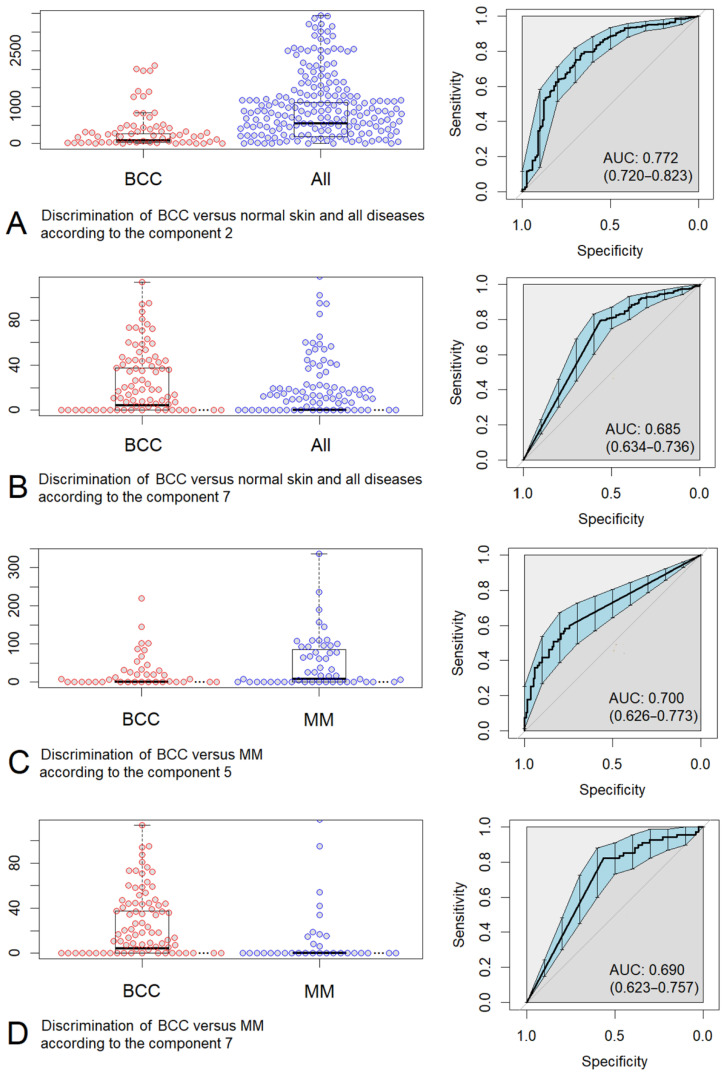
Box-plot diagrams and ROC curves corresponding to several cases of discrimination: (**A**) basal cell carcinoma (BCC) versus normal skin and all diseases (All) according to the MCR Component 2 (melanin), (**B**) basal cell carcinoma (BCC) versus normal skin and all diseases (All) according to the MCR Component 7 (proteins, lipids, nature moisturizing factor, melanin), (**C**) basal cell carcinoma (BCC) versus malignant melanoma (MM) according to the MCR Component 5 (proteins, lipids, nature moisturizing factor), (**D**) basal cell carcinoma (BCC) versus malignant melanoma (MM) according to the MCR Component 7 (proteins, lipids, nature moisturizing factor, melanin).

**Figure 7 sensors-22-09588-f007:**
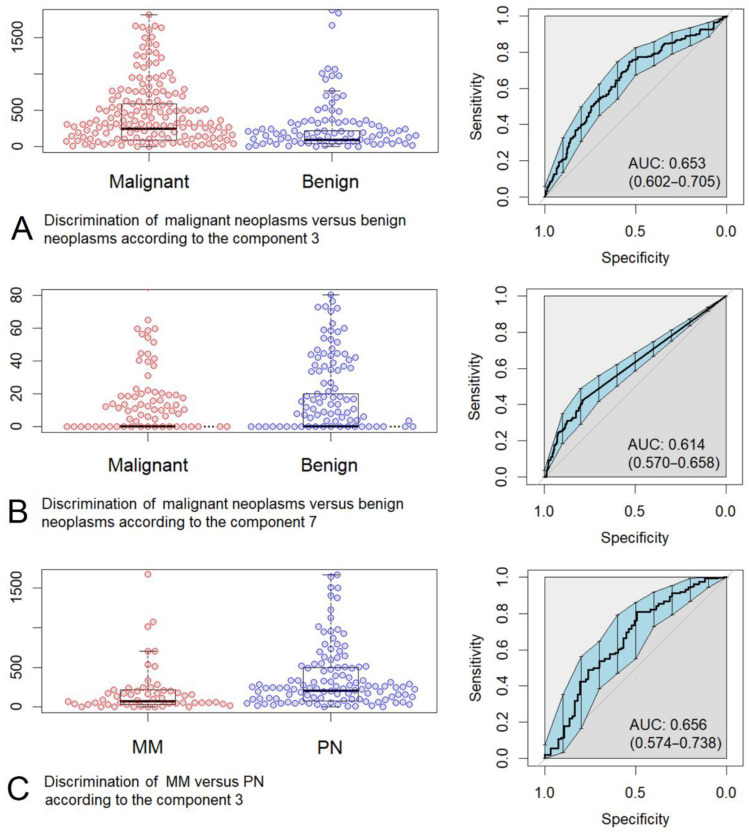
Box-plot diagrams and ROC curves corresponding to several cases of discrimination: (**A**) malignant neoplasms (basal cell carcinoma + malignant melanoma) versus benign neoplasms (keratosis + pigmented nevus) according to the MCR Component 3 (proteins), (**B**) malignant neoplasms (basal cell carcinoma + malignant melanoma) versus benign neoplasms (keratosis + pigmented nevus) according to the MCR Component 7 (proteins, lipids, nature moisturizing factor, melanin), (**C**) malignant melanoma (MM) versus pigmented nevus (PN) according to the MCR Component 3 (proteins).

**Figure 8 sensors-22-09588-f008:**
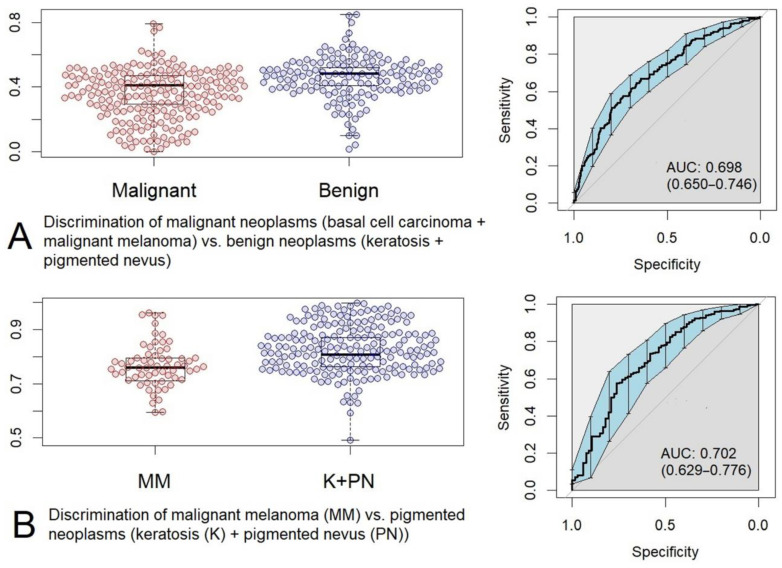
Box-plot diagrams and ROC-curves of predicted values obtained as a result of logistic regression: (**A**) malignant neoplasms (basal cell carcinoma + malignant melanoma) versus benign neoplasms (keratosis + pigmented nevus), (**B**) malignant melanoma (MM) versus pigmented neoplasms (keratosis + pigmented nevus).

**Figure 9 sensors-22-09588-f009:**
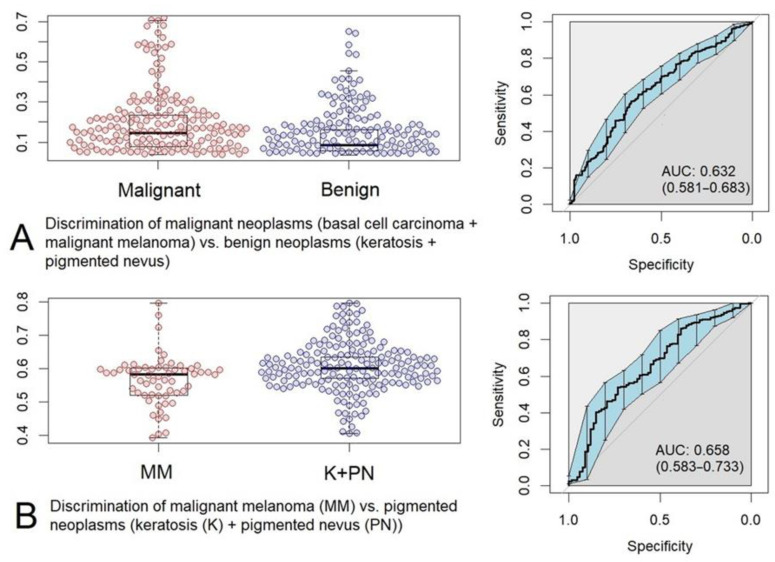
Box-plot diagrams and ROC-curves of the predicted values obtained as a result of PLS-DA: (**A**) malignant neoplasms (basal cell carcinoma + malignant melanoma) versus benign neoplasms (keratosis + pigmented nevus), (**B**) malignant melanoma (MM) versus pigmented neoplasms (keratosis + pigmented nevus).

**Table 1 sensors-22-09588-t001:** The significance values (*p*-values) for component concentration calculated at the first step of logistic regression. The *p*-values for the coefficients indicate whether these relationships are statistically significant. If the *p*-value for a variable is less than the significance level of 0.05, this variable is statistically significant. Bold type in the table indicates cases of binary logistic regression.

	Malignant vs. Benign	MM vs. K + PN
Component 1 (optical system contribution)	0.004	0.101
Component 2 (melanin)	0.449	0.243
Component 3 (proteins)	0.034	0.805
Component 4 (water)	0.005	0.002
Component 5 (proteins, lipids, nature moisturizing factor)	0.753	0.710
Component 6 (proteins, lipids)	0.183	0.499
Component 7 (proteins, lipids, nature moisturizing factor, melanin)	0.043	0.246
Component 8 (proteins, water)	0.017	0.568

## Data Availability

Not applicable.

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
