# Peer review of "Multivariate Curve Resolution Alternating Least Squares Analysis of In Vivo Skin Raman Spectra"

_sensors, 2022, doi:10.3390/s22249588_

Round 1
Reviewer 1 Report
Classifying skin cancers with Raman spectroscopy based on the concentration of known skin constituents remain a challenge, because of the difficulty to separate the Raman features of multiple biomarkers. In this manuscript, the authors aim to address this challenge by using multivariate curve resolution alternating least squares (MCR-ALS) method. The authors applied the MCR-ALS method to a large dataset collected by a portable Raman probe in a clinical setting, and then used the unmixed components for classification. The results demonstrate the promises of proposed method in classifying the skin neoplasms while capturing their biophysical differences. The manuscript is worth publishing after addressing the following concerns:
1. The authors stated that the SNR of the in vivo Raman spectra used in this study was low. The definition of “low” seems vague. It would be recommended to perform a quantitative analysis of the SNR.
2. The authors stated that it was difficult to acquire high SNR Raman spectra in a clinical setting, and that the spectroscopic setup had a lower SNR compared to the Raman microscope. This statement was only partially true because SNR was determined by multiple factors, including the laser power, the integration time, and the collection efficiency of the detector. As an example, the reference below used a Raman probe for clinical setting but still reached a high-SNR (> 50), as shown in Figure 4 of the reference below:
Desroches, J., Jermyn, M., Mok, K., Lemieux-Leduc, C., Mercier, J., St-Arnaud, K., Urmey, K., Guiot, M.C., Marple, E., Petrecca, K. and Leblond, F., 2015. Characterization of a Raman spectroscopy probe system for intraoperative brain tissue classification. Biomedical optics express, 6(7), pp.2380-2397.
3. Why was the wavenumber range below 1114 1/cm not used? For examples, Raman bands were assigned to nucleus at 835 1/cm and 1093 1/cm. The sensitivity of the method in detecting nucleus may be compromised.
4. In figure 1a, the raw spectrum of pigmented nevus is not shown.
5. In figure 3, it is interesting to see a sinusoidal pattern in component #5 and some other components. Have the authors tried optical response calibration to reduce this pattern? For example, by measuring a relative intensity correction standard for Raman spectroscopy with 785 nm excitation.
6. The authors mentioned that the skin spectra in Ref [34] also had a contribution of optics; however, the optics in Ref [34] did not show a sinusoidal pattern. Is it possible that, there is other source for this sinusoidal pattern, such as the etaloning effect from the back-thinned CCD of the spectrometer? This could also be possibly corrected by optical response calibration.
7. Please exam over the manuscript to avoid redundancy. For instance, Line 290 – 305 and Line 465 – 480 are exactly the same. Line 520 – 523 and Line 683 – 686 are also the same. Please delete one of them.
8. The reviewer has two comments for this statement in Conclusion: “In this study, we were unable to unmix them into separate components. The use of more accurate equipment for recording Raman spectra, for example, a Raman microscope, could make it possible to achieve better results.” Firstly, the term “accurate” should be replaced with “localized measurement”. It is not that the portable probe is inaccurate, but just that it averages skin spectrum within a larger sampling volume. Secondly, the authors should notice that one advantage of using Raman microscope is that MCR-ALS can yield 2D microscopic imaging (such as Ref [21]), thus can build a direct relation between the skin constituents and the unmixed components by comparing with the histology. Similarly, to calibrate and confirm the true composition of the skin with a portable Raman probe, it is worthwhile to try the method in this paper (see Figure 1):
Nguyen, H.T., Zhang, Y., Moy, A.J., Feng, X., Sebastian, K.R., Reichenberg, J.S., Fox, M.C., Markey, M.K. and Tunnell, J.W., 2021, July. Characterization of Ex Vivo Nonmelanoma Skin Tissue Using Raman Spectroscopy. In Photonics (Vol. 8, No. 7, p. 282). MDPI.
Author Response
Dear Reviewer,
We are grateful for your critical reading and constructive suggestions. Please find our detailed replies below:
- The authors stated that the SNR of the in vivo Raman spectra used in this study was low. The definition of “low” seems vague. It would be recommended to perform a quantitative analysis of the SNR.
A quantitative analysis of the SNR under various recording conditions was carried out in our previous paper (see Ref 7). We have added the value of the SNR coefficient to the text explicitly in section 1. Introduction and subsection 2.1. Experimental data:
“In this research, we use Raman portable setup [7]. The methodology for the evaluation the effectiveness of a portable setup and SNR estimation are described in Ref [7].”
“In this case, the SNR of the recorded Raman spectra is 3 in accordance with the methodology for SNR estimation described in Ref [7].”
We also provided a reference to an article where the calculation methodology is described and all calculations are performed:
Khristoforova, Y. A.; Bratchenko, I. A.; Myakinin, O. O.; Artemyev, D. N.; Moryatov, A. A.; Orlov, A. E.; Kozlov, S. V.; Zakharov, V. P. Portable spectroscopic system for in vivo skin neoplasms diagnostics by Raman and autofluorescence analysis. Journal of biophotonics 2019, 12(4), e201800400.
- The authors stated that it was difficult to acquire high SNR Raman spectra in a clinical setting, and that the spectroscopic setup had a lower SNR compared to the Raman microscope. This statement was only partially true because SNR was determined by multiple factors, including the laser power, the integration time, and the collection efficiency of the detector. As an example, the reference below used a Raman probe for clinical setting but still reached a high-SNR (> 50), as shown in Figure 4 of the reference below:
Desroches, J., Jermyn, M., Mok, K., Lemieux-Leduc, C., Mercier, J., St-Arnaud, K., Urmey, K., Guiot, M.C., Marple, E., Petrecca, K. and Leblond, F., 2015. Characterization of a Raman spectroscopy probe system for intraoperative brain tissue classification. Biomedical optics express, 6(7), pp.2380-2397.
The SNR of the recorded Raman spectra depends on the type of spectrometer, the detector cooling temperature, the signal accumulation time, etc. In the reference, it was revealed that there is a linear relationship between SNR and temperature (see Figure 5 in the reference). The high SNR was achieved due to the strong cooling of the detector (the temperature of the camera was set to −40°C). However, it is not a handheld setup, as we see from the reference. In contrast, we utilized a portable handheld spectroscopic setup with a CCD detector operating at -15 °C.
We have added a little discussion on this topic in section 1. Introduction and a reference to the article you provided:
“A SNR is determined by multiple factors, including the laser power, the detector cooling temperature, the signal accumulation time, etc. It was revealed that there is a linear relationship between SNR and temperature (see Ref [24]). Although it is possible to increase the SNR due to the strong cooling of the detector, however, the portability will be lost.”
- Why was the wavenumber range below 1114 1/cm not used? For examples, Raman bands were assigned to nucleus at 835 1/cm and 1093 1/cm. The sensitivity of the method in detecting nucleus may be compromised.
As we use portable setup (i.e. low SNR level) the autofluorescence signal overlaps week Raman peaks in the spectral region from 300 to 1200 1/cm for tissue samples with high melanin content, which make the determination of Raman peaks below 1200 1/cm unstable. Therefore, only autofluorescence skin features can be analyzed in this region. The same conclusions were reported in the paper:
- Mahadevan-Jansen, R. Richards-Kortum, J. Biomed. Opt. 1996, 1(1), 31. https://doi.org/10.1117/12.227815.
In opposite, due to the autofluorescence exponentially decrease it was observed a smaller autofluorescence contribution in the range from 1200 to 1800 1/cm, which allows one extracting tissue Raman peaks from registered signal. Thus, tissue Raman peaks are important features of the 1200 to 1800 1/cm spectral range (see Ref [7]).
- In figure 1a, the raw spectrum of pigmented nevus is not shown.
Changes applied. The figure 1 was replaced.
- In figure 3, it is interesting to see a sinusoidal pattern in component #5 and some other components. Have the authors tried optical response calibration to reduce this pattern? For example, by measuring a relative intensity correction standard for Raman spectroscopy with 785 nm excitation.
In this study, we did not try optical response calibration to reduce this sinusoidal pattern and did not use it to pre-process the Raman spectra and thereby increase the signal-to-noise ratio. The reduction of the sinusoidal pattern will be in the focus of our further research.
- The authors mentioned that the skin spectra in Ref [34] also had a contribution of optics; however, the optics in Ref [34] did not show a sinusoidal pattern. Is it possible that, there is other source for this sinusoidal pattern, such as the etaloning effect from the back-thinned CCD of the spectrometer? This could also be possibly corrected by optical response calibration.
Yes, you are absolutely right. We do not know the exact reason for observed sinusoidal pattern as we not investigate this feature in details, it is planned for our further research. But the authors of paper [35] observe similar effect (dependence of optics) and they associate this effect with an etaloning effect from the back-thinned CCD of the spectrometer. The text has been changed to clarify this issue more clearly:
“A component representing the contribution of optics (but without a sinusoidal pattern) was obtained in Ref [35] where the authors applied a non-negative matrix factorization (NNMF) algorithm to the in vivo Raman spectra of upper epidermis. It can be assumed that the source for the sinusoidal pattern is an etaloning effect from the back-thinned CCD of the spectrometer. The reason for this sinusoidal effect and ways to reduce it will be the focus of our further research.”
- Please exam over the manuscript to avoid redundancy. For instance, Line 290 – 305 and Line 465 – 480 are exactly the same. Line 520 – 523 and Line 683 – 686 are also the same. Please delete one of them.
Changes applied. The lines deleted.
- The reviewer has two comments for this statement in Conclusion: “In this study, we were unable to unmix them into separate components. The use of more accurate equipment for recording Raman spectra, for example, a Raman microscope, could make it possible to achieve better results.” Firstly, the term “accurate” should be replaced with “localized measurement”. It is not that the portable probe is inaccurate, but just that it averages skin spectrum within a larger sampling volume. Secondly, the authors should notice that one advantage of using Raman microscope is that MCR-ALS can yield 2D microscopic imaging (such as Ref [21]), thus can build a direct relation between the skin constituents and the unmixed components by comparing with the histology. Similarly, to calibrate and confirm the true composition of the skin with a portable Raman probe, it is worthwhile to try the method in this paper (see Figure 1):
Nguyen, H.T., Zhang, Y., Moy, A.J., Feng, X., Sebastian, K.R., Reichenberg, J.S., Fox, M.C., Markey, M.K. and Tunnell, J.W., 2021, July. Characterization of Ex Vivo Nonmelanoma Skin Tissue Using Raman Spectroscopy. In Photonics (Vol. 8, No. 7, p. 282). MDPI.
Changes applied. We also have added a note about using Raman microscope in section 4. Conclusion:
“The use of more localized measurement equipment for recording Raman spectra, for example, a Raman microscope, could make it possible to achieve better results.
Besides, one advantage of using Raman microscope is that in this case we can yield 2D microscopic imaging (see Ref [21]) and build a direct relation between the skin constituents and the unmixed components by comparing with the histology.”

Reviewer 2 Report
The content of this paper was to report the use of the multivariate curve resolution alternating least squares (MCR-ALS) method for the analysis of Raman spectroscopy signals with skin tissues in vivo. It provided useful information and a novel theory of the data analysis of MCR-ALS method.
However, the structure of the article is not organized, and the content of the paper is easy to confuse.
1. Please enhance the quality of figures, such as Figures 4 and 5.
2. All statistical methods need to be explained. For example, which regression analysis was used, and why select this statistical method? The logistic regression and ROC AUC did not be illustrated.
3. In “Table 1. The significance values for component concentration calculated at the first step of logistic regression”, eight components are listed and provide a clear picture of the study in the .paper. Please make a detailed table. For each component, the conditions of the data distribution curves, data processing method, and data analysis technique (MCR-ALS, logistic regression, or ROC AUC) are illustrated. This table could make the paper be more readable.
Author Response
Dear Reviewer,
We are grateful for your critical reading and constructive suggestions. Please find our detailed replies below:
- Please enhance the quality of figures, such as Figures 4 and 5.
Changes applied. Figures 4 and 5 replaced.
- All statistical methods need to be explained. For example, which regression analysis was used, and why select this statistical method? The logistic regression and ROC AUC did not be illustrated.
We used a binary regression analysis. A description of logistic regression, binary logistic regression, PLS-DA and how to read a box-plot, ROC, and ROC AUC is added in subsection 2.3. Analysis of results:
“A box plot shows the distribution of quantitative data in a way that facilitates comparisons between variables or across levels of a categorical variable. The box shows the quartiles of the dataset while the whiskers extend to show the rest of the distribution, except for points that are determined to be outliers.
ROC curve illustrates the diagnostic ability of a binary classifier system as its discrimination threshold is varied. The ROC curve is created by plotting the true positive rate (TPR) against the false positive rate (FPR) at various threshold settings. The true-positive rate is also known as sensitivity. The false-positive rate is also known as probability and can be calculated as (1 − specificity).
Area under the ROC curve (AUC) provides an aggregate value of how well the model correctly classifies the cases with all possible cutoff values. AUC values range between 0.5 and 1, where an area of 0.5 means that the model predicts which outcomes will be 1 or 0 no better than flipping a coin, and an area of 1 means that the model predicts perfectly.”
“This type of statistical model is often used for classification in medicine and biology. Binary logistic regression estimates the probability of an event occurring, such as melanoma or no melanoma, based on a given dataset of independent variables. In other words, the goal of binary logistic regression is to simply classify observations as one of the two possible outcomes that the model describes.”
“Partial least squares discriminant analysis (PLS-DA) is one of the most widely used classification techniques in many fields, which generates large amount of data such as spectroscopy. PLS-DA is a supervised clustering or classification method, used to optimize separation between different groups of samples, which is completed by linking two data matrices X (i.e., raw data) and Y (i.e., classes, or diagnoses). This method aims to maximize the covariance between the independent variables X and the corresponding dependent variable Y of highly multidimensional data by finding a linear subspace of the explanatory variables. The new subspace permits the prediction of the Y variable based on a reduced number of factors (PLS components).”
We use such methods because they are widely used in the field of medical diagnostics, which allows us to compare our results with those of other researchers.
- In “Table 1. The significance values for component concentration calculated at the first step of logistic regression”, eight components are listed and provide a clear picture of the study in the paper. Please make a detailed table. For each component, the conditions of the data distribution curves, data processing method, and data analysis technique (MCR-ALS, logistic regression, or ROC AUC) are illustrated. This table could make the paper be more readable.
We suppose that making a table with data processing method, and data analysis technique for each component is redundant, because for each component, we used the same data processing and analysis methods, which are consistently described in the text. To make the article clearer and more understandable, we also have added a more extended description of some of the steps in the research.
Table 1. The significance values (p-values) for component concentration calculated at the first step of logistic regression. The p-values for the coefficients indicate whether these relationships are statistically significant. If the p-value for a variable is less than the significance level of 0.05, this variable is statistically significant. Bold type in the table indicates cases of binary logistic regression.
|
|
Malignant vs. benign |
MM vs. K+PN |
|
Component 1 (optical system contribution) |
0.004 |
0.101 |
|
Component 2 (melanin) |
0.449 |
0.243 |
|
Component 3 (proteins) |
0.034 |
0.805 |
|
Component 4 (water) |
0.005 |
0.002 |
|
Component 5 (proteins, lipids, nature moisturizing factor) |
0.753 |
0.710 |
|
Component 6 (proteins, lipids) |
0.183 |
0.499 |
|
Component 7 (proteins, lipids, nature moisturizing factor, melanin) |
0.043 |
0.246 |
|
Component 8 (proteins, water) |
0.017 |
0.568 |
As can be seen from Table 1, in the case of the classification of malignant vs. benign tumors, significant components are 1, 3, 4, 7, and 8. Earlier, we assumed that these components correspond to optical system contribution, proteins, water, lipids, nature moisturizing factor, and melanin. Feng et al. [21] also report about the changes in the concentration of collagen during disease development; for instance, in the case of BCC, the relative concentration of collagen decreases, while in the case of PN, it increases. It should be noted, that Component 2 is expected to become less significant due to the presence of melanin in both groups.
In the case of classification of MM vs. pigmented neoplasms (K+PN), Component 3, which reflects the contribution of proteins, becomes less important. Indeed, Feng et al. [21] reports that with the development of K and MM, the share of collagen changes in approximately the same way. Components 6, 7 and 8 are also become less significant, because they also represent the contribution of proteins.

Round 2
Reviewer 1 Report
The authors have addressed all my concerns. I thus recommend the publication of this manuscript.
Author Response
Dear Reviewer,
We are grateful for your appreciation of the article. The replied content was added to the text of the mauscript.
Reviewer 2 Report
The authors provided detailed replies to the previous comment. All questions have been replied to appropriately.
Please add these replied contents to the revision paper. It will enhance the quality of this paper.
Author Response
Dear Reviewer,
We are grateful for your appreciation of the article. The replied content was added to the text of the mauscript.
See the attached file for the details.
